# Is radiographic lumbar spinal stenosis associated with the quality of life?: The Wakayama Spine Study

**Satoshi Arita[1], Yuyu Ishimoto[1]\*, Hiroshi Hashizume[1], Keiji Nagata[1], Shigeyuki Muraki[2], Hiroyuki Oka[3], Masanari Takami[1], Shunji Tsutsui[1], Hiroshi Iwasaki[1], Yasutsugu Yukawa[1], Toru Akune[4], Hiroshi Kawaguchi[5], Sakae Tanaka[6], Kozo Nakamura[4], Munehito Yoshida[7], Noriko Yoshimura[2◉], Hiroshi Yamada[1◉], Consortium[¶]**

**1** Department of Orthopaedic Surgery, Wakayama Medical University, Wakayama City, Wakayama, Japan, **2** Department of Preventive Medicine for Locomotive Organ Disorders, 22nd Century Medical and Research Centre, Faculty of Medicine, The University of Tokyo, Bunkyo-ku, Tokyo, Japan, **3** Department of Medical Research and Management for Musculoskeletal Pain, 22nd Century Medical & Research Centre, The University of Tokyo Hospital, Bunkyo-ku, Tokyo, Japan, **4** Rehabilitation Services Bureau, National Rehabilitation Centre for Persons with Disabilities, Tokorozawa, Saitama, Japan, **5** Department of Orthopaedics and Spine, Tokyo Neurological Centre, Minato-ku, Tokyo, Japan, **6** Department of Orthopaedic Surgery, Faculty of Medicine, The University of Tokyo Hospital, Bunkyo-ku, Tokyo, Japan, **7** Department of Orthopaedic Surgery, Sumiya Orthopaedic Hospital, Wakayama City, Wakayama, Japan

◉ These authors contributed equally to this work.
¶ Membership of the Consortium is listed in the Acknowledgments.
\* yuyu.ishimoto@gmail.com

**Data Availability Statement:** All relevant data are within the manuscript and its Supporting information files.

## Abstract

### Objectives

This prospective study aimed to determine the association between radiographic lumbar spinal stenosis (LSS) and the quality of life (QOL) in the general Japanese population.

### Methods

The severity of radiographic LSS was qualitatively graded on axial magnetic resonance images as follows: no stenosis, mild stenosis with ≤1/3 narrowing, moderate stenosis with a narrowing between 1/3 and 2/3, and severe stenosis with > 2/3 narrowing. Patients less than 40 years of age and those who had undergone previous lumbar spine surgery were excluded from the study. The Oswestry Disability Index (ODI), which includes 10 sections, was used to assess the QOL. One-way analysis of variance was performed to determine the statistical relationship between radiographic LSS and ODI. Further, logistic regression analysis adjusted for gender, age, and body mass index was performed to detect the relationship.

### Results

Complete data were available for 907 patients (300 men and 607 women; mean age, 67.3 ±12.4 years). The prevalence of severe, moderate, and non-mild/non-radiographic were 30%, 48%, and 22%, respectively. In addition, the mean values of ODI in each group were 12.9%, 13.1%, and 11.7%, respectively, and there was no statistically significant difference

**Funding:** This work was supported by H-25-Choujyu-007 (Director, NY), H25-Nanchitou (Men)-005 (Director, ST), and 201417014A (Director, NY) from the Ministry of Health, Labour and Welfare, a Grant-in-Aid for Scientific Research (C 26861206) of JSPS KAKENHI grant. And Collaborating Research with NSF 08033011–00262 (Director, NY) from the Ministry of Education, Culture, Sports, Science and Technology in Japan. This study also was supported by grants from the Japan Osteoporosis Society (NY, HO, and TA), a grant from JA Kyosai Research Institute (HO), Japan Society for the Promotion of Science, Grants-in-Aid for Scientific Research (KAKENHI) Research C (17K10937) (MT), (20K09439)(YI), a Grant from the Japanese Orthopaedics and Traumatology Foundation, Inc (No. 287) (MT). The funders had no role in study design, data collection and analysis, decision to publish, or preparation of the manuscript.

**Competing interests:** The authors have declared that no competing interests exist.

between the three groups in logistic analysis (P = 0.55). In addition, no significant differences in any section of the ODI were observed among the groups. However, severe radiographic LSS was associated with low back pain in the "severe" group as determined by logistic analysis adjusted for gender, age, and body mass index (odds ratio: 1.53, confidence interval: 1.13–2.07) compared with the non-severe group.

## Conclusion

In this general population study, severe radiographic LSS was associated with low back pain (LBP), but did not affect ODI.

## Introduction

Lumbar spinal stenosis (LSS) is a painful degenerative disorder [1–8], with an estimated prevalence of 6% to 47%, depending on the diagnostic criteria and study subjects [9–12]. It is characterised by neurogenic claudication, which consists of lower limb pain and neurological symptoms that are exacerbated by walking. LSS is the most common reason for spine surgery in patients aged over 65 years [13], with a current estimated 2-year cost of $4 billion in the United States [14, 15]. Given the ageing population, both the prevalence and economic burden of LSS are expected to increase [13–19]. Therefore, there is an urgent need for a clear solution to this economic burden. LSS is also one of the three major diseases constituting the 'locomotive syndrome', as advocated by Nakamura [20] in 2000. However, the extent to which it negatively affects the lives of the general population remains unclear. To the best of our knowledge, the association between radiographic LSS and the quality of life (QOL) has not been investigated in the Japanese general population. In this study, we aimed to determine the association of the relationship between radiographic LSS and QOL in a population-based cohort.

## Materials and methods

### Study design

The Wakayama Spine Study (WSS) prospectively assessed a sub-cohort from the Research on Osteoarthritis/Osteoporosis Against Disability (ROAD) study, a large-scale, prospective study of bone and joint disease among population-based cohorts in Japan [21–24].

### Participants

The ROAD study's database included the baseline clinical and genetic information of 3040 patients (1061 men, 1979 women) with a mean age of 70.6 years (range: 23–95 years). Individuals listed in the resident registrations in the following three communities were recruited for the study: (i) an urban region in Itabashi, Tokyo; (ii) a mountainous region in Hidakagawa, Wakayama; (iii) a coastal region in Taiji, Wakayama. All the participants provided written informed consent before the commencement of the study, which was conducted with the approval of the ethics committees of the University of Tokyo and the Tokyo Metropolitan Institute of Gerontology. The participants completed an interviewer-administered questionnaire consisting of 400 questions, including those on lifestyle, and underwent anthropometric measurements and assessments of physical performance. Blood and urine samples were collected for biochemical and genetic examination. The ankle-brachial index of all the participants (OMRON Co. Kyoto, Japan) was also measured. The ROAD study team made a second

visit to the mountainous region of Hidakagawa and the coastal region of Taiji between 2008 and 2010. Of the inhabitants who participated in this second visit, 1,063 volunteers were recruited for MRI. Fifty-two of these declined to attend the examination, and the remaining 1,011 were registered in the Wakayama Spine Study. All participants provided their written, informed consent for the MRI examination. Participants who had sensitive implanted devices (such as a pacemaker) or other disqualifiers were excluded. In total of 977 participants underwent a lumbar spine MRI in a mobile MRI unit. Ten participants who underwent a previous lumbar surgery for LSS and 29 participants aged <40 years were excluded from the study because LSS is a degenerative disease. Complete MRI and ODI data were available for 907 participants (300 men and 607 women), who were included in this study (Fig 1).

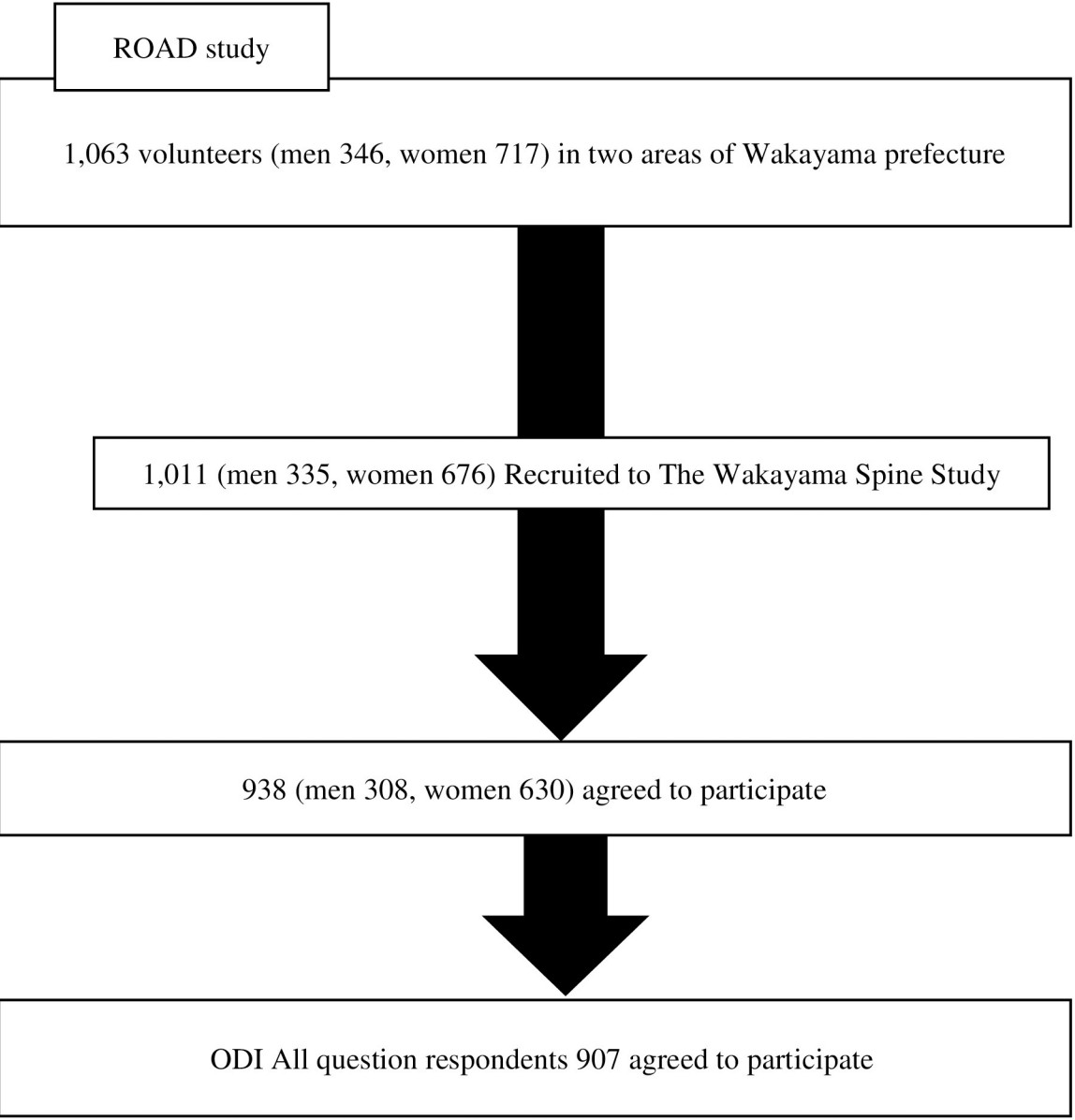

**Fig 1. Flow diagram depicting participants recruited to the Wakayama Spine Study from the ROAD study.**

## MRI

All the subjects underwent total spinal MRI using a pre-defined standard protocol in a mobile unit (Excelart 1.5 T; Toshiba; Tokyo, Japan). MRI was not performed in patients with a cardiac pacemaker, claustrophobia, or other relevant contraindications. The participants were positioned supine, and those with rounded backs were positioned with triangular pillows under their head and knees. The imaging protocol was as follows: sagittal T2-weighted fast spin-echo (FSE; repetition time [TR]: 4000 ms/echo; echo time [TE]: 120 ms; field of view [FOV]: 300 × 20 mm), and axial T2-weighted FSE (TR: 4000 ms/echo; TE: 120 ms; FOV: 180 × 180 mm). Axial images were taken at each lumbar intervertebral level (L1/2-L5/S1) parallel to the vertebral endplates.

## Assessment of radiographic LSS

Despite the severity of symptoms that can result from LSS, there is no consensus on to how to define LSS radiologically using MRI scanning [25], although many approaches have been suggested [26]. For the current study, the severity of LSS on MRI scans was assessed qualitatively by an experienced spine surgeon (YI) following the methodology of Suri et al. [27]. The severity of central canal stenosis was qualitatively graded on the axial images as follows: no spinal stenosis; mild spinal stenosis, with a maximum of 1/3 narrowing; moderate spinal stenosis, with narrowing between 1/3 and 2/3; severe spinal stenosis, with more than 2/3 narrowing (Fig 2). To confirm the reliability of this method, the observer reassessed a random sample of 50 of the MRI scans after a period of one month, blinded to the original rating, and achieved excellent intra-observer reliability with a kappa of 0.82 (95% CI: 0.77–0.86). Inter-observer variability was measured between the study observer and another experienced spine surgeon

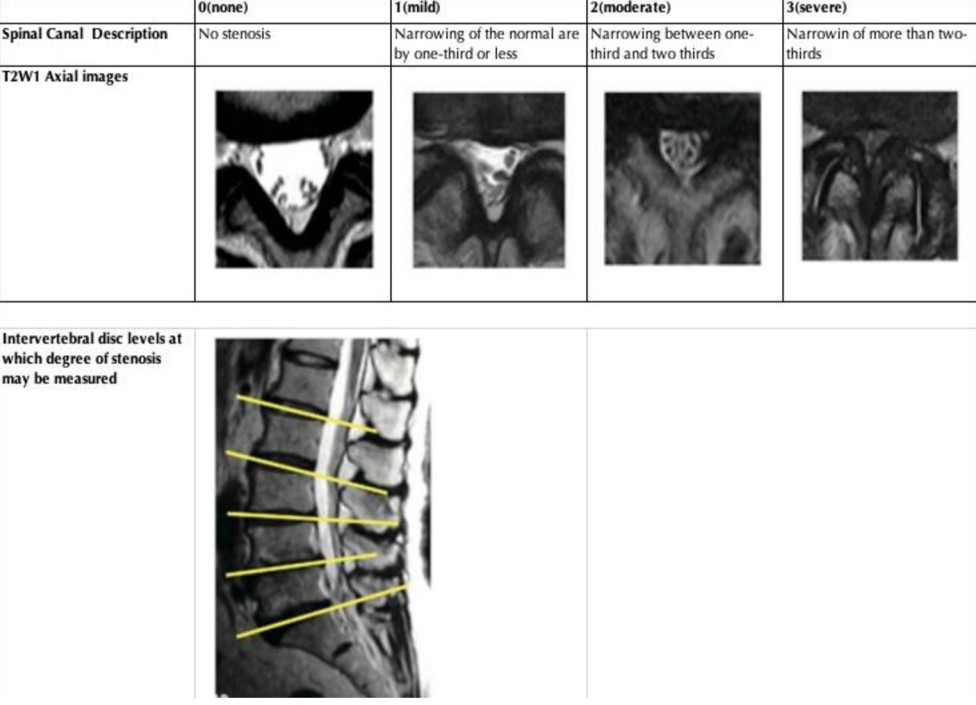

**Fig 2. Qualitative central stenosis grading [24].** dol.org/10.1002/ajlm.22957.

(KN) for a different sample of 50 MRI scans, achieving a kappa of 0.77 (95% CI: 0.73–0.82) for agreement. None of the included MRI scans were found to have LSS caused by tumour, inflammatory, or traumatic pathologies.

## Questionnaire, interview, and anthropometric measurements

THE participants completed a 400-item interviewer-administered questionnaire that assessed their lifestyle characteristics, such as occupation, smoking habits, alcohol consumption, family history, medical history, physical activity, reproductive variables, and health-related QOL. Current smokers were defined as those who smoked, regardless of the number of pack-years, while never and former smokers were classified as non-smokers. Current habitual alcohol consumption was defined as alcohol consumption regardless of the amount; never and former drinkers were classified as non-drinkers. Anthropometric measurements included height and weight, and the body mass index (BMI) was then calculated [BMI; weight (kg)/height$^2$ (m$^2$)]. Medical details regarding the participants' systemic, local, and mental status were obtained by experienced orthopaedists.

## ODI

Clinicians and researchers use the ODI, an index derived from the Oswestry Low Back Pain (LBP) Questionnaire [28–30], to quantify the level of disability due to LBP. This patient questionnaire includes several topics, including pain intensity, the ability to walk, sit, stand, care for oneself, travel, sexual function, lifting, social life, and sleep quality.

The subjects were asked to select the statement that most closely resembled their symptoms. The index scores ranged from 0 to 100, with '0' indicating no disability and '100' indicating the most severe disability.

## Statistical analysis

All statistical analyses were performed using JMP version 14 (SAS Institute Japan, Tokyo, Japan). The association of the average ODI with radiographic LSS severity was examined using one-way analysis of variance. Similarly, the same test was used to examine the association between radiographic LSS severity and the average ODI for each question. The relationship between radiographic LSS and LBP was examined using the logistic regression analysis adjusted for age, sex, and BMI.

## Results

Table 1 summarises the characteristics of the 907 participants (300 men and 607 women; mean age: 67.3 years, range: 40–93 years), including their age and anthropometric measurements.

The mean age did not differ significantly between men and women, but the BMI was significantly lower in women than in men. The average ODI for all participants was 12.8% (Fig 3), while for each radiographic LSS severity, it was (severe, moderate, non-mild) was 12.9%, 13.1%, and 11.7%, respectively, with no statistically significant difference between the three groups (P = 0.55) (Fig 4). In addition, according to the severity of radiographic LSS (severe, moderate, non-mild), the mean percentages for each question were as follows: question 1 (0.92%, 0.98%, and 1.0%, respectively), question 2 (0.32%, 0.37%, and 0.35%, respectively), question 3 (0.79%, 1.01%, and 0.95%, respectively), question 4 (0.6%, 0.58%, and 0.61%, respectively), question 5 (0.64%, 0.71%, and 0.68%, respectively), question 6 (0.87%, 0.95%, and 0.89%, respectively), question 7 (0.22%, 0.20%, and 0.26%, respectively), question 9

**Table 1. Characteristics of participants.**

|  | All | Male | Female |
|---|---|---|---|
| N | 907 | 300 | 607 |
| Age (years) | 67.3 (±12.4) | 68.4 (±12.6) | 66.8 (±12.4) |
| Age group |  |  |  |
| <49 | 95 | 26 | 69 |
| 50–59 | 169 | 57 | 112 |
| 60–69 | 214 | 63 | 151 |
| 70–79 | 247 | 85 | 162 |
| ≧80 | 182 | 69 | 113 |
| BMI (kg/m$^2$) | 23.3 (±3.6) | 23.7 (±3.3) | 23.1 (±3.7) |
| Height (cm) | 155.7 (±9.3) | 164.4 (±6.9) | 151.4 (±7.2) |
| Weight (kg) | 56.7 (±11.4) | 64.3 (±11.3) | 53.0 (±9.4) |
| LBP | 371 | 111 | 251 |

Values are mean ± SD unless otherwise indicated.

BMI, body mass index; LBP, low back pain.

(0.52%, 0.86%, and 0.97%, respectively), and question 10 (0.39%, 0.67%, and 0.48%, respectively) (Fig 5).

Question 8 on sex life was excluded in this study. There were no significant differences in any of the questions among the groups.

The severity of radiographic LSS (severe, moderate, non-mild), the mean percentages for each question were as follows: question 1 (0.92%, 0.98%, and 1.0%, respectively), question 2 (0.32%, 0.37%, and 0.35%, respectively), question 3 (0.79%, 1.01%, and 0.95%, respectively), question 4 (0.6%, 0.58%, and 0.61%, respectively), question 5 (0.64%, 0.71%, and 0.68%, respectively), question 6 (0.87%, 0.95%, and 0.89%, respectively), question 7 (0.22%, 0.20%, and 0.26%, respectively), question 9 (0.52%, 0.86%, and 0.97%, respectively), and question 10 (0.39%, 0.67%, and 0.48%, respectively).

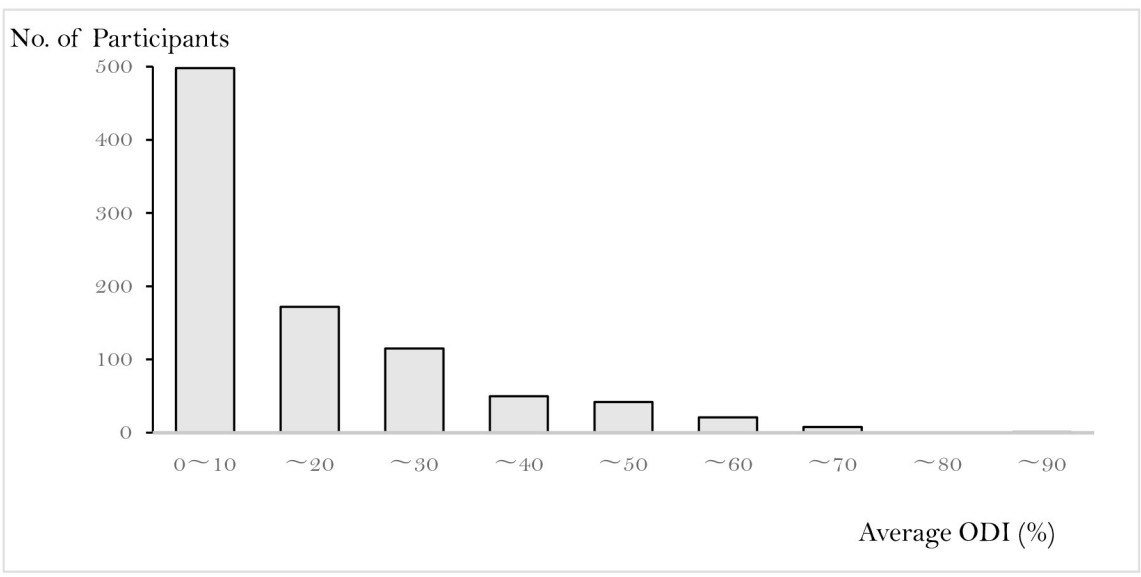

**Fig 3. Distribution of ODI score.** The participants' average ODI is 12.8%.

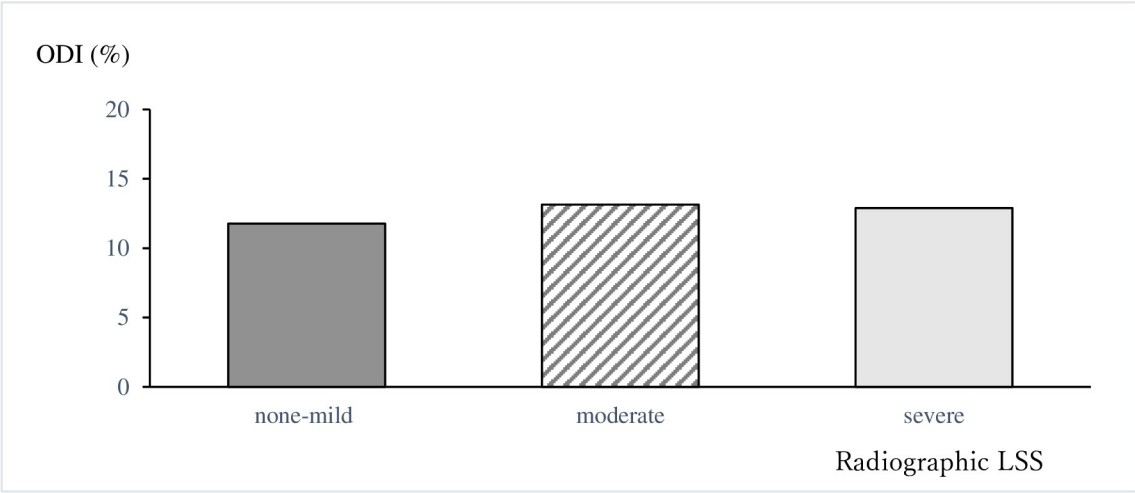

**Fig 4. The average ODI for each radiographic LSS group.** The average ODI for each radiographic LSS group (severe, moderate, mild/none) is 12.9%, 13.1%, and 11.7%, respectively, which are not statistically different (P = 0.55).

There was no significant difference in the ODI scores of any of the questions among the three groups.

## Discussion

This study found the average ODI of radiographic LSS to be 12.9% in the severe group, 13.1% in the moderate group, and 11.7% in the non-mild group, with no statistically significant difference among them (P = 0.55) (Fig 4).

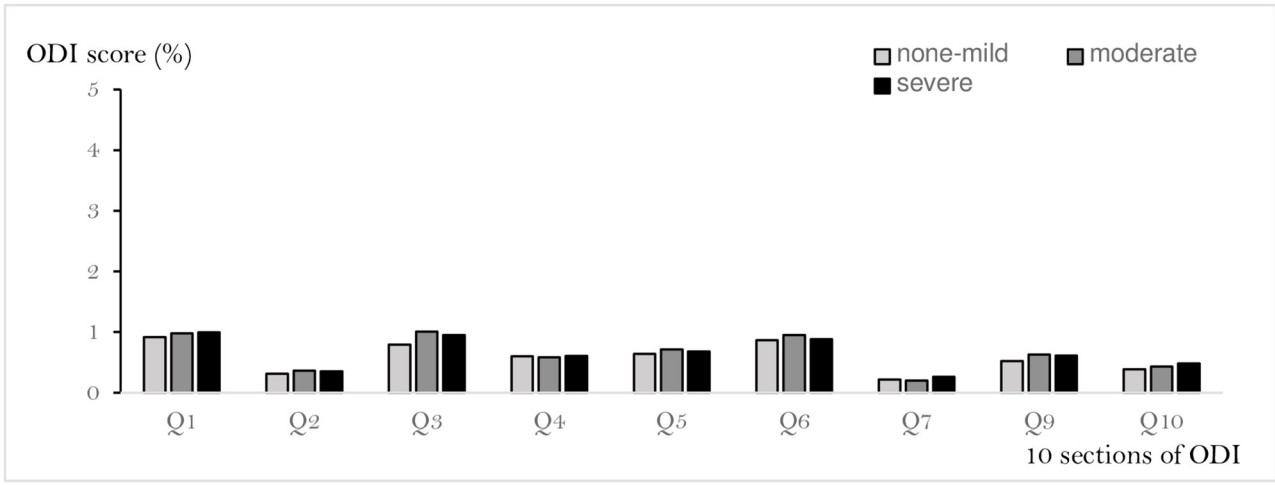

**Fig 5. The score of each 10 sections of ODI vs radiographic LSS.** This figure shows the association of the severity of radiographic LSS (severe, moderate, non-mild) with average score (%) for each question were as follows: question 1 (0.92%, 0.98%, and 1.0%, respectively), question 2 (0.32%, 0.37%, and 0.35%, respectively), question 3 (0.79%, 1.01%, and 0.95%, respectively), question 4 (0.6%, 0.58%, and 0.61%, respectively), question 5 (0.64%, 0.71%, and 0.68%, respectively), question 6 (0.87%, 0.95%, and 0.89%, respectively), question 7 (0.22%, 0.20%, and 0.26%, respectively), question 9 (0.52%, 0.86%, and 0.97%, respectively), and question 10 (0.39%, 0.67%, and 0.48%, respectively). There was no significant difference in the ODI scores of any of the questions by the severity of radiographic LSS. We also excluded the Q8 associated sexual function.

In addition, there was no significant difference in the ODI scores of any of the questions among the three groups.

There have been varying reports on the relationship between radiographic LSS and the QOL using LSS patients. Kanno et al. [31] reported that the dural sac cross-sectional area on MRI correlated highly with walking distance and with the Japanese Orthopaedic Association score in 88 outpatients with LSS. Ogikubo et al. [32] reported that a smaller preoperative minimum sac cross-sectional area was associated with lower walking distance, back pain, and QOL, while Borden et al. [33] suggested that 21% of asymptomatic volunteers aged > 60 years had LSS. Conversely, Lohman et al. [34] found no association between the cross-sectional area and clinical symptoms in patients with LBP and clinical suspicion of LSS. Ishimoto [10] clarified that about 80% of the participants had radiographic LSS severity greater than mild stenosis, but only less than 20% of those with severe stenosis were symptomatic. Thus, it seems impossible to clarify the cause of clinical symptoms using static imaging alone. In their 10-year follow-up study, Minamide et al. [35] stated that the condition of only 30% of patients with LSS worsened, suggesting that these patients maintain their activities of daily living and QOL by walking in a hunched posture, putting their hands on their knees, or walking with a wheelbarrow. Furthermore, using a bicycle may also be possible because lumbar extension significantly decreases the canal area, whereas flexion has the opposite effect. These ingenious ways may also have helped the participants of our study in maintaining their QOL. Among the three radiographic LSS groups, no significant differences in the ODI scores were noted. There was no significant difference in the score of the first question about pain [None-mild: 1.09 (±0.93), Moderate: 1.01 (±0.91), Severe: 0.82 (±0.96)]. Iwahashi [36] showed that in the in WSS, a narrow cross-sectional area (less than 1/4 of the normal dural sac area) was associated with LBP after adjustment for age, sex, and BMI. Our results also found a relationship between the severity of radiographic LSS and LBP, but not with the questions on QOL regarding pain (Fig 6). In our cohort, since only 10% of the individuals were symptomatic, our participants may not have been severely affected by nerve compression. Radiographic LSS may be secondary to lumbar osteoarthritis, including spondylolisthesis, scoliosis, and disc degeneration. LBP due to degeneration is often position-dependent [37]; therefore, despite LBP, by avoiding painful positions, such patients may live without a significant decline in their QOL.

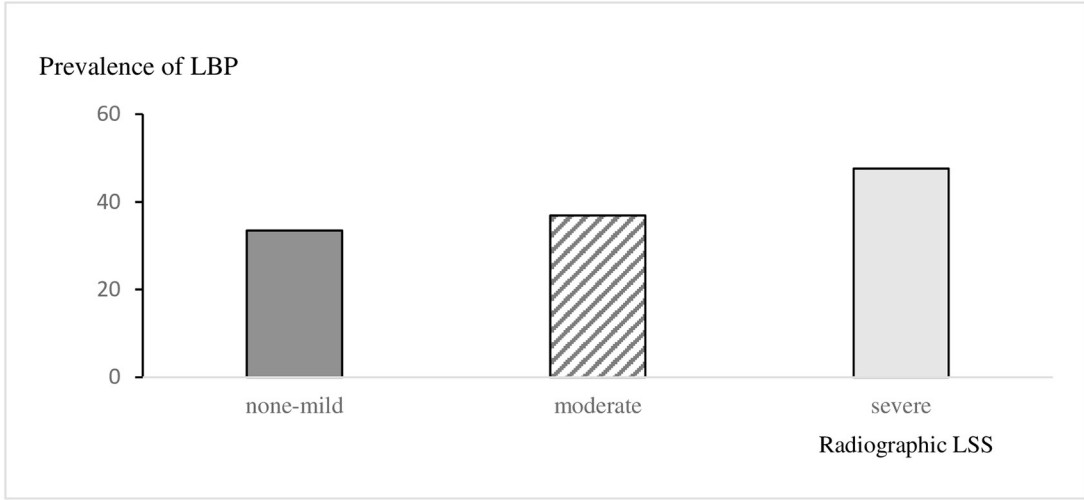

**Fig 6. LBP vs. radiographic LSS.** LBP prevalence increases 0.5-fold as the severity of the disease progresses from non-mild to severe, while it increases 0.6-fold with a moderate-to-severe progression.

LBP prevalence increases 0.5-fold as the severity of the disease progresses from non-mild to severe, while it increases 0.6-fold with a moderate-to-severe progression.

## Limitations

Despite its findings, there are several limitations to this study. First, this was a cross-sectional study, so causal attributions could not be made. Second, the participants in this study were sampled from the general population, but not randomly. We investigated their representativeness by comparing the study population with Japan's general population as a key risk factor for osteoarthritis and BMI. We found that the mean BMI of the participants did not differ significantly from that of the general Japanese population (males: 23.71 (±3.41) vs. 23.95 (±2.64) kg/m$^2$, respectively; women: 23.06 (±3.42) vs. 23.50 (±3.69) kg/m$^2$, respectively). However, the study participants reported a lower prevalence of smoking and alcohol consumption than the general Japanese population, suggesting that our subjects might have had healthier lifestyles. This may limit the generalizability of our findings. We also could not rule out selection bias, as volunteers needed to be sufficiently healthy to participate and undergo spinal radiographs. This may have limited the possible involvement of elderly institutionalised adults, since LBP is a common cause of impaired mobility in older people, which may lead to institutionalisation. Finally, the ODI in this study reflects only the LBP-related QO and not necessarily the overall QOL. In addition, we did not use a scale such as the visual analog scale to assess low back pain.

Nevertheless, this study was the first to evaluate the association between radiographic LSS and ODI in the general population. The strength of this study was that all the MRI scans were assessed by a highly trained orthopaedic surgeon (YI) with high reliability, including inter-observer and intra-observer studies with a sample of 5% of the MRI scans. Since the WSS is a longitudinal study, future results will help to clarify the causal relationships of the factors involved. In addition, as described in the North American Spine Society guidelines, such a prospective study evaluating the changes in the severity of imaging and clinical findings over time among untreated patients with moderate LSS will provide Level I evidence for the natural history of the disease. This study is the first step in this direction.

## Conclusions

This study investigated the relationship between radiographic LSS and ODI in the general Japanese population and found that radiographic LSS was associated with LBP, but not with QOL. Our results suggest that radiographic LSS can coexist with the patients' daily living. However, it is also true that some of the mild cases may become more severe and lead to surgery. Further longitudinal surveys of The Wakayama Spine Study will help to further clarify the aggravating factor for LBP and QOL.

## Supporting information

**S1 File.**
(XLSX)

## Acknowledgments

The authors would like to thank Mrs. Tomoko Takijiri and other members of the Public Office in Hidakagawa Town, and Mrs. Tamako Tsutsumi, Mrs. Kanami Maeda, and other members of the Public Office in Taiji Town for their assistance in locating and scheduling participants for examinations.

No benefits in any form have been or will be received from a commercial party related directly or indirectly to the subject of this manuscript.

## Author Contributions

**Conceptualization:** Satoshi Arita, Yuyu Ishimoto, Hiroshi Hashizume, Toru Akune, Hiroshi Kawaguchi, Kozo Nakamura, Munehito Yoshida, Noriko Yoshimura.

**Data curation:** Satoshi Arita, Shigeyuki Muraki, Munehito Yoshida, Noriko Yoshimura.

**Formal analysis:** Satoshi Arita, Yuyu Ishimoto, Hiroshi Hashizume, Hiroyuki Oka.

**Funding acquisition:** Yuyu Ishimoto, Hiroshi Hashizume, Hiroyuki Oka, Toru Akune, Sakae Tanaka, Munehito Yoshida, Noriko Yoshimura, Hiroshi Yamada.

**Investigation:** Satoshi Arita, Yuyu Ishimoto, Hiroshi Hashizume, Keiji Nagata, Shigeyuki Muraki, Hiroyuki Oka, Masanari Takami, Shunji Tsutsui, Hiroshi Iwasaki, Yasutsugu Yukawa, Sakae Tanaka, Munehito Yoshida, Noriko Yoshimura, Hiroshi Yamada.

**Methodology:** Satoshi Arita, Yuyu Ishimoto, Hiroshi Hashizume, Keiji Nagata, Hiroyuki Oka, Yasutsugu Yukawa.

**Project administration:** Satoshi Arita, Toru Akune, Hiroshi Kawaguchi, Kozo Nakamura, Noriko Yoshimura.

**Resources:** Satoshi Arita.

**Supervision:** Hiroshi Hashizume, Yasutsugu Yukawa, Toru Akune, Hiroshi Kawaguchi, Sakae Tanaka, Kozo Nakamura, Munehito Yoshida, Hiroshi Yamada.

**Validation:** Satoshi Arita, Yuyu Ishimoto, Hiroshi Hashizume, Masanari Takami, Shunji Tsutsui.

**Visualization:** Satoshi Arita.

**Writing – original draft:** Satoshi Arita, Yuyu Ishimoto.

**Writing – review & editing:** Satoshi Arita, Yuyu Ishimoto, Hiroshi Hashizume, Shigeyuki Muraki, Hiroyuki Oka, Shunji Tsutsui, Hiroshi Iwasaki, Yasutsugu Yukawa, Sakae Tanaka, Munehito Yoshida, Noriko Yoshimura.

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
