## [Decision Letter · Decision Letter 0]

3 Jan 2022

PONE-D-21-31516A large-scale, population-based cohort study in Japan to determine the effect of lumbar spinal stenosis seen on MRI on the quality of life: The Wakayama Spine StudyPLOS ONE

Dear Dr. Ishimoto,

Thank you for submitting your manuscript to PLOS ONE. After careful consideration, we feel that it has merit but does not fully meet PLOS ONE’s publication criteria as it currently stands. Therefore, we invite you to submit a revised version of the manuscript that addresses the points raised during the review process.

We look forward to receiving your revised manuscript.

Kind regards,

Fatih Özden, PhD

Academic Editor

PLOS ONE

Journal Requirements:

Additional Editor Comments (if provided):

Three reviewers have positive opinions on this paper. I also find the topic and aim useful. If you suitably manage the minor revisions, this manuscript would have an essential contribution to the literature. Your revised paper will be sent to the reviewers. Best Regards

Reviewers' comments:

Reviewer's Responses to Questions

**Comments to the Author**

1. Is the manuscript technically sound, and do the data support the conclusions?

Reviewer #1: Yes

Reviewer #2: Yes

Reviewer #3: Yes

2. Has the statistical analysis been performed appropriately and rigorously? 

Reviewer #1: Yes

Reviewer #2: Yes

Reviewer #3: I Don't Know

3. Have the authors made all data underlying the findings in their manuscript fully available?

Reviewer #1: Yes

Reviewer #2: Yes

Reviewer #3: Yes

4. Is the manuscript presented in an intelligible fashion and written in standard English?

Reviewer #1: Yes

Reviewer #2: Yes

Reviewer #3: Yes

5. Review Comments to the Author

Reviewer #1: Report on

A large-scale, population-based cohort study in Japan to determine the effect of lumbar spinal stenosis seen on MRI on the quality of life: The Wakayama Spine Study

or

Effect of lumbar spinal stenosis seen on MRI on the quality of life

General Comments:

Lumbar spinal stenosis is a narrowing of the spinal canal in the lower part of the patient's back. The manuscript aimed to aimed to determine the association between radiographic lumbar spinal stenosis (LSS) and the quality of life (QOL) in the general Japanese population.

907 patients were clustered into 4 groups no, mild, moderate, and severe stenosis. The study was prospective which is a very good advantage.

Exclusion Criteria:

1. Patients younger than 40 years.

2. Patients who had undergone previously to lumber spine surgery.

In this general population study, severe radiographic LSS was associated with LBP, but did not affect ODI.

The topic is interesting and is of relevance to the readers of the journal.

Introduction

The introduction was well organized and fully explained.

Methodology

1. Description of the experimental procedures was concise and contained detailed descriptions of well-established procedures.

2. The number of patients was sufficient.

3. Inclusion and Exclusion criteria seem logical.

Results and Discussion

1. The results were represented by nice and informative radiographic images showing LSS.

2. The study findings were adequately discussed.

Decision

The study is interesting. Therefore, I recommend it for publication as it is.

Reviewer #2: Thanks for inviting me to review the manuscript entitled “A large-scale, population-based cohort study in Japan to determine the effect of lumbar spinal stenosis seen on MRI on the quality of life: The Wakayama Spine Study ”. I am glad to have an opportunity to give some comments on this work.

1. In general, I think this is a nice work done by a large group of researchers and doctors, who conducted a quite large-scale study. I fully understand this is not easy and appreciate the hardworking.

2. The title is wordy, so it could be more concise.

3. In abstract, the font size is not consistent. And in methods, logistic regression analysis was not mentioned but in results, it was used in fact.

4. In the Introduction, the words "radiographic LSS and the quality of life (QOL) has not been investigated in the Japanese general population.. Therefore, we aimed to assess the effect of radiographic LSS on the QOL using mobile magnetic resonance imaging (MRI) and the Oswestry Disability Index (ODI) in them" is not scientific, and more importantly, in the discussion part (paragraph 3) described that "there are vary reports on the relationship between radiographic LSS and QOL.."

5. How many cases were excluded and the reason?

6. As this is a large-scale clinical study, some useful clinical relevance in Conclusion would be more attractive.

Reviewer #3: The manuscript titled: A large-scale, population-based cohort study in Japan to determine the effect of lumbar

spinal stenosis seen on MRI on the quality of life: The Wakayama Spine Study is good and interesting.

Some comments were found.

Abstract:

Page 6 Line 91: The prevalence of severe, moderate, and non-mild/non-radiographic ..........This sentence incomplete

Line 100: LBP2: Abbreviation should be written in complete form when mentioned for the first time in the manuscript

Materials and Methods:

Page 11 Line 182-187: Number of patient is incorrect (It should be938 patient). You should explain this.

Fig. 5. The figure should be corrected.

6. PLOS authors have the option to publish the peer review history of their article (what does this mean?). If published, this will include your full peer review and any attached files.

Reviewer #1: **Yes: **Amr Abd-Elghany

Reviewer #2: No

Reviewer #3: No

---

## [Author Response · Author response to Decision Letter 0]

14 Jan 2022

Reviewer #1: 

>Thank you very much for your comment. 

Reviewer #2: 

2. The title is wordy, so it could be more concise.

>Thank you for your advice. I have changed the title as you said.

“Is radiographic lumbar spinal stenosis associated with the quality of life?: The Wakayama Spine Study”

3. In abstract, the font size is not consistent. And in methods, logistic regression analysis was not mentioned but in results, it was used in fact.

>I corrected the font size in abstract. I also added the sentence about logistic regression analysis into the methods. 

4. In the Introduction, the words "radiographic LSS and the quality of life (QOL) has not been investigated in the Japanese general population.. Therefore, we aimed to assess the effect of radiographic LSS on the QOL using mobile magnetic resonance imaging (MRI) and the Oswestry Disability Index (ODI) in them" is not scientific, and more importantly, in the discussion part (paragraph 3) described that "there are vary reports on the relationship between radiographic LSS and QOL.."

>Thank you for your comment. In fact, there have been several studies on the relationship between radiographic LSS and the QOL, though, most their subjects were patients who were outpatients or underwent surgery. We could not find the study on the association of LSS with QOL using general people.　I deleted above sentence and added another sentence in the introduction. 

5. How many cases were excluded and the reason?

> Thank you for your comment. I added the some sentences in the methods for clarity. Firstly, 1,063 volunteers were recruited for MRI from the second visit ROAD study. Fifty-two of these declined to attend the examination, and the remaining 1,011 were registered in the Wakayama Spine Study. Participants who had sensitive implanted devices (such as a pacemaker) or other disqualifiers were excluded. In A total of 977 participants underwent a lumbar spine MRI in a mobile MRI unit. Ten participants who underwent a previous lumbar surgery for LSS and 29 participants aged <40 years were excluded from the study because LSS is a degenerative disease.

6. As this is a large-scale clinical study, some useful clinical relevance in Conclusion would be more attractive.

>Thank you for your comment. I added the below sentences in Conclusion.

However, it is also true that some of the mild cases may become more severe and lead to surgery. Further longitudinal surveys of The Wakayama Spine Study will help to further clarify the aggravating factor for LBP and QOL.

Reviewer #3: 

Abstract:

Page 6 Line 91: The prevalence of severe, moderate, and non-mild/non-radiographic ..........This sentence incomplete

>Thank you very much. I completed the sentence.

Line 100: LBP2: Abbreviation should be written in complete form when mentioned for the first time in the manuscript

>Thank you very much. I corrected as you mentioned.

Materials and Methods:

Page 11 Line 182-187: Number of patient is incorrect (It should be938 patient). You should explain this.

> Your point is right. I corrected the number and added some sentences for clarity.

Fig. 5. The figure should be corrected.

>Thank you for your advice. I corrected the title of Fig.5 and the names for the vertical and horizontal axes to make it easier to understand. In addition, I changed the color of bars. (The more severe stenosis, the darker the color)

I also added the last sentence about Q8 associated sexual function.

---

## [Decision Letter · Decision Letter 1]

31 Jan 2022

Is radiographic lumbar spinal stenosis associated with the quality of life?: The Wakayama Spine Study

PONE-D-21-31516R1

Dear Dr. Ishimoto,

We’re pleased to inform you that your manuscript has been judged scientifically suitable for publication and will be formally accepted for publication once it meets all outstanding technical requirements.

Kind regards,

Fatih Özden, PhD

Academic Editor

PLOS ONE

Reviewers' comments:

Reviewer's Responses to Questions

**Comments to the Author**

1. If the authors have adequately addressed your comments raised in a previous round of review and you feel that this manuscript is now acceptable for publication, you may indicate that here to bypass the “Comments to the Author” section, enter your conflict of interest statement in the “Confidential to Editor” section, and submit your "Accept" recommendation.

Reviewer #1: All comments have been addressed

Reviewer #2: All comments have been addressed

Reviewer #3: All comments have been addressed

2. Is the manuscript technically sound, and do the data support the conclusions?

Reviewer #1: Yes

Reviewer #2: Yes

Reviewer #3: Yes

3. Has the statistical analysis been performed appropriately and rigorously? 

Reviewer #1: Yes

Reviewer #2: Yes

Reviewer #3: Yes

4. Have the authors made all data underlying the findings in their manuscript fully available?

Reviewer #1: Yes

Reviewer #2: Yes

Reviewer #3: Yes

5. Is the manuscript presented in an intelligible fashion and written in standard English?

Reviewer #1: Yes

Reviewer #2: Yes

Reviewer #3: Yes

6. Review Comments to the Author

Reviewer #1: The authors responded to all the reviewers comments. I have no further comments to authors and recommend it to be accepted.

Reviewer #2: The questions are well addressed and there are no further comments. It is time to accept this artwork!

Reviewer #3: The manuscript is interesting and good written and discussed.

Abstract is good written

Introduction is good written

Materaials and results are good written and illustrated

Discussion is good written

7. PLOS authors have the option to publish the peer review history of their article (what does this mean?). If published, this will include your full peer review and any attached files.

Reviewer #1: **Yes: **Amr Abd-Elghany

Reviewer #2: No

Reviewer #3: No

---

## [Editor Report · Acceptance letter]

8 Feb 2022

PONE-D-21-31516R1 

Is radiographic lumbar spinal stenosis associated with the quality of life?: The Wakayama Spine Study 

Dear Dr. Ishimoto:

I'm pleased to inform you that your manuscript has been deemed suitable for publication in PLOS ONE. Congratulations! Your manuscript is now with our production department. 

Kind regards, 

on behalf of

Dr. Fatih Özden 

Academic Editor

PLOS ONE